# G-Protein-Coupled Receptors Mediate Modulations of Cell Viability and Drug Sensitivity by Aberrantly Expressed Recoverin 3 within A549 Cells

**DOI:** 10.3390/ijms24010771

**Published:** 2023-01-01

**Authors:** Hanae Ichioka, Yoshihiko Hirohashi, Tatsuya Sato, Masato Furuhashi, Megumi Watanabe, Yosuke Ida, Fumihito Hikage, Toshihiko Torigoe, Hiroshi Ohguro

**Affiliations:** 1Departments of Ophthalmology, School of Medicine, Sapporo Medical University, Sapporo 060-8556, Japan; 2Departments of Pathology, School of Medicine, Sapporo Medical University, Sapporo 060-8556, Japan; 3Departments of Cardiovascular, Renal and Metabolic Medicine, Sapporo Medical University, Sapporo 060-8556, Japan; 4Departments of Cellular Physiology and Signal Transduction, Sapporo Medical University, Sapporo 060-8556, Japan

**Keywords:** 3D spheroid culture, melanoma, RNA sequencing, Gene Ontology (GO) enrichment analysis, ingenuity pathway analysis (IPA), G-protein-coupled receptors, recoverin

## Abstract

To elucidate the currently unknown molecular mechanisms responsible for the aberrant expression of recoverin (Rec) within cancerous cells, we examined two-dimensional (2D) and three-dimensional (3D) cultures of Rec-negative lung adenocarcinoma A549 cells which had been transfected with a plasmid containing human recoverin cDNA (A549 Rec) or an empty plasmid as a mock control (A549 MOCK). Using these cells, we measured cytotoxicity by several anti-tumor agents (2D), cellular metabolism including mitochondrial and glycolytic functions by a Seahorse bio-analyzer (2D), the physical properties, size and stiffness of the 3D spheroids, trypsin sensitivities (2D and 3D), and RNA sequencing analysis (2D). Compared with the A549 MOCK, the A549 Rec cells showed (1) more sensitivity toward anti-tumor agents (2D) and a 0.25% solution of trypsin (3D); (2) a metabolic shift from glycolysis to oxidative phosphorylation; and (3) the formation of larger and stiffer 3D spheroids. RNA sequencing analysis and bioinformatic analyses of the differentially expressed genes (DEGs) using Gene Ontology (GO) enrichment analysis suggested that aberrantly expressed Rec is most likely associated with several canonical pathways including G-protein-coupled receptor (GPCR)-mediated signaling and signaling by the cAMP response element binding protein (CREB). The findings reported here indicate that the aberrantly expressed Rec-induced modulation of the cell viability and drug sensitivity may be GPCR mediated.

## 1. Introduction

It is well known that paraneoplastic syndromes (PNSs) are associated with malignant tumors with no evidence of tumor invasion toward the nervous system [1]. It is possible that this is caused by autoimmune effects against aberrantly expressed neuron-specific antigens that are produced within tumor cells and that these immunological responses lead to neuronal cell degeneration [1,2,3]. Among the PNSs, cancer-associated retinopathy (CAR) has been identified as a major ocular PNS [4,5,6] and is frequently found in patients with small-cell carcinomas of the lung and other malignant tumors [7,8,9]. It is known that CAR is characterized as a form of retinitis pigmentosa-like photoreceptor degeneration with the following clinical manifestations: photopsia, progressive visual field loss with a ring scotoma and attenuated retinal arterioles, in addition to the abnormal a- and b-waves of an electroretinogram (ERG) [10]. Possible causative autoantigens related to CAR include several retina-specific and non-specific molecules that have been identified [11,12]. Among these, a photoreceptor-specific Ca^2+^ binding protein, recoverin (Rec) [13,14] has been the most extensively characterized in terms of both their physiological and pathological roles. Thus, Rec has been reported to play a pivotal role in the light and dark adaptation of photoceptor cells by regulating rhodopsin phosphorylation and that this process is calcium-dependent [15]. Pathologically, Rec itself has been identified as a potent immunogenic molecule and in fact, has been reported to cause experimental uveoretinitis in rats [16]. The following issues have been postulated to occur during CAR pathogenesis: (1) serum anti-Rec antibodies are generated by unknown mechanisms in some patients with malignant tumors; and (2) once those anti-Rec antibodies are produced, they penetrate photoreceptor cells, thereby inhibiting physiological Rec functions, thus leading to the death of photoreceptor cells [17,18]. Therefore, during the initial phase of CAR pathogenesis, the aberrant expression of Rec serving as the common antigen between tumor cells and the retina should be a key mechanism for causing a relatively rare ocular PNS, CAR [19,20,21]. However, our previous studies revealed that such aberrant expressions of Rec were identified in more than 50% of various types of cancer cells [22] as well as in established cancerous cell lines [18,23]. Therefore, this unexpectedly high aberrant Rec expression in cancerous cells rationally suggest that Rec may have some specific roles within cancerous cells. In fact, our previous studies also found that the artificial expression of the human Rec in A549 lung adenocarcinoma cells without Rec expression caused a significant decrease in cell proliferation [18,22,23], and a substantial increase in the sensitivities toward anti-cancer drugs [22]. These collective observations support the relatively favorable prognosis of the underlying malignant tumors in patients with CAR [10]. However, to date, only a very limited amount of information regarding the molecular pathophysiological roles of the aberrantly expressed Rec within the cancerous cells is available.

Therefore, to obtain additional insights into the roles of Rec within cancerous cells, Rec-positive and -negative A549 cells were prepared, and, using two-dimensional (2D) and three-dimensional (3D) cultures, these cells were subjected to the following analyses: (1) real-time analysis of cellular metabolism to evaluate biological activities, and (2) RNA sequencing analysis to elucidate possible biological signaling mechanisms responsible for causing such aberrantly expressed Rec induced effects that are described above.

## 2. Results

To study the effects of the aberrant expression of Rec within A549 lung adenoma cells, human recoverin cDNA was transfected and a positive expression was confirmed within A549 Rec, but not A549 WT nor A549 MOCK by a qPCR analysis (Appendix A), and these calls were subjected to 2D and 3D cell cultures. Initially, cytotoxicity by several anti-tumor agents, and real-time cellular metabolic functions of the 2D-cultured cells were evaluated (Figure 1). The cytotoxicity induced by DTX in the 2D-cultured A549 Rec was increased as compared with control MOCK cells as observed in our previous studies [22,23]. However, in contrast, significant differences in the anti-tumor drug sensitivities by CRDCA or PEM were not observed between A549 MOCK and A549 Rec, suggesting that aberrantly expressed Rec may influence specific but not all mechanisms by anti-tumor drug, i.e., the inhibition of cellular mitosis mechanisms by DTX [24] rather than modulations of DNA-related metabolisms by CRDCA [25] and PEM [26].

In addition, the cellular metabolic characteristics of A549 Mock and A549 Rec cells were assessed. As shown in Figure 2, the A549 Rec cells showed a significant increase in Mitochondrial Spare Reserve Capacity, a significant decrease in the rate of ATP production during glycolysis, and a significant increase in the mitoOCR/glycoPER ratio compared with the A549 Mock cells, suggesting that Rec induces metabolic changes between glycolysis and oxidative phosphorylation (OXPHOS).

To study the aberrant expression of Rec in terms of the spatial oncogenesis of the A549 cells, we prepared A549 Rec and A549 Mock 3D spheroids and their appearance and physical properties, size and stiffness were compared. As shown in Figure 3A,B, the A549 3D spheroids had a disc-shaped appearance, and grew smaller with their maturation during the 6-day culture, as has been observed for other non-cancerous cells such as human orbital fibroblast [27], 3T3-L1 preadipocytes [28], and human trabecular meshwork cells [29]. However, the mean sizes of A549 Rec 3D spheroids were significantly larger than the A549 WT or MOCK 3D spheroids (Figure 3C). Furthermore, as demonstrated in Figure 4, the physical stiffness of the A549 WT or Rec 3D spheroids were significantly higher as compared with the A549 MOCK 3D spheroids. Such Rec-induced effects toward the spatial architecture of the 3D A549 spheroids were also confirmed by the difference in their sensitivities to trypsin. Thus, although the course of digestion of the 2D-cultured cells with 0.025% trypsin was not significantly different between A549 MOC and A549 Rec, that of the 3D A549 Rec spheroids was much faster as compared with 3D A549 MOCK spheroids (Figure 5). Therefore, these collective findings indicate that the aberrant expression of Rec exerted a significant influence on the physical properties, size and stiffness, and 3D spatial architecture of the 3D spheroids in addition to the cytotoxicity induced by the anti-tumor agents and real-time cellular metabolism.

To elucidate the currently unidentified mechanisms responsible for inducing such characteristic features between A549 Rec and A549 Mock, RNA sequencing analyses were performed (GSE218942). As shown in MA and volcano plots (Figure 6A,B), 31 significantly up-regulated and 57 down-regulated differentially expressed genes (DEGs) were identified for A549 Rec as compared with A549 Mock with a significance level of <0.05 (FDR) and an absolute fold-change ≥ 2 was identified (a list of the up-regulated and down-regulated genes is shown in Appendix A). To estimate the possible functional roles of the above DEGs that were detected, we conducted a GO enrichment analysis and an Ingenuity Pathway Analysis (IPA) and (Qiagen, Redwood City, CA, USA). As the top three significant canonical pathways, G-protein-coupled receptor (GPCR) signaling (−log (*p*-value) = 6.3), CREB signaling (−log (*p*-value) = 5.8), Gia mediated signaling (−log (*p*-value) = 5.2) were identified based on the detected DEGs (Appendix A).

## 3. Discussion

In the current study, RNA sequencing in conjugation with GO analysis and IPA led to a rational assumption that aberrantly expressed Rec may affect GPCR-related signaling because it is known that Rec functions as a GPCR kinase 1 called “rhodopsin kinase” in a Ca^2+^-dependent manner [15]. In fact, in our previous study employing an immunoprecipitation approach using anti-human Rec mAb and a Rec-positive cell line, HEK293 cells (human embryonic kidney cell line), we found that GPCR kinases 2, 5 and 6, and caveolin-1 were co-precipitated with Rec [30]. An immunocytochemistry analysis also revealed that the immunoreactivities toward Rec, GRKs and caveolin-1 had merged. In fact, the consensus sequence motif for caleolin-1 binding [31] is present within the amino acid sequence of Rec [30], and GPCR kinase 2, 5 and 6 homologous to GPCR kinase 1 [32] are distributed within caveola and associated with caveolin-1 [33]. Caveolae are omega-shaped invaginations of the plasma membrane and are comprised of glycosphingolipid- and cholesterol-enriched components, on which a variety of signaling events can take place [34,35,36]. It appears that caveolins are functionally associated with several important signal-transduction-related proteins; GPCRs including β2-adrenergic, m2-muscarinic, B2-bradykinin, cholecystokinin, ETA-endothelin and angiotensin II receptor, G proteins including Gsα, Giα, Goα and Gqα, and various downstream effectors including GPCR kinases, adenylate cyclase, PKC, and NO synthase, in addition to mitogen signaling cascades and related factors including the epidermal growth factor, platelet derived growth factor, insulin and c-ErbB2 receptors including c-Src, Fyn, Erk-2, and Ras [37,38,39,40,41,42,43]. Therefore, these collective findings quite reasonably suggest that Rec which is aberrantly expressed in A549 cells affects several cellular functions and structures including sensitivity to anti-tumor agents, cellular metabolism and the physical properties of 3D spheroids, via GPCR-related signaling, CREB signaling and Gia-mediated signaling mechanisms.

However, current hypotheses remain speculative at present because of the following study limitations that would need to be investigated for them to be overcome: (1) Our current experimental system expressed high levels of Rec after transfection. Therefore, in order to understand the pathophysiological aspects of the aberrantly expressed Rec within cancerous cells, it would be better to use cells with cells that express lower levels of Rec or that naturally and aberrantly express Rec. Although we have very limited knowledge concerning the phenotypes of such lower Rec expressing tumor cells, Yamaji et al. established a naturally Rec expressed small-cell lung carcinoma (SCLC) cell line, designated MN-1112, from a patient with SCLC who showed CAR syndrome. In the characterization of their biological aspects demonstrated that morphologic and immunocytochemical features, and efficacies of cell growth, production of tumor makers, and carcinogenesis toward nude mice of MN-1112 cells were quite similar to those of the classic type of SCLC cell lines [20]. However, in our previous studies that involved the immunostaining of Rec from surgically obtained cancerous tissues the expression was higher in the case of earlier clinical cancerous stages [22,44] which rationally support the current data showing higher sensitivities of A549 Rec against anti-tumor drugs (Figure 1). (2) The relationship between the analysis of the RNAseq/pathway and the difference in cellular metabolic states based on Seahorse Bioanalyzer measurements between A549 Mock and A549 Rec is not fully understood at this time. However, some mitochondria-related genes were identified within the DEGs (Appendix A). In addition, in Ant1 (Adenine nucleotide translocator 1) knockout mice, a significantly up-regulated retinal Rec expression was found to be expressed within the inner mitochondrial membrane [45,46], in which oxidative phosphorylation (OXPHOS) was decreased as compared with WT [47,48]. These observations are consistent with our current results. (3) As shown in Figure 4 and Figure 5, the stiffness of the A549 Mock 3D spheroids was lower, but their trypsin resistance was higher as compared with A549 Rec. Although these two physical aspects may not be completely understood, we speculate that more complicated underlying mechanisms including plasticity as well as elasticity may be involved in the case of the stiffness measurements of these 3D spheroids. In fact, upon the administration of trypsin, we observed the earlier swelling of the 3D A549 Rec spheroids as compared with 3D A549 Mock, but both 3D spheroids remained intact for periods of up to 12 h. In addition, genes related to several integral and anchoring components of membranes were identified among the DEGs. Therefore, additional functional and morphological investigations that involve modulating the currently identified genes as well as several possible related biological pathways will be required as our next project.

## 4. Materials and Methods

### 4.1. 2D and 3D Cultures of A549 Lung Adenoma Cells Transfected with or without Human Recoverin cDNA

A549 lung adenoma cells were obtained from the American Type Culture Collection (Manassas, VA, USA). Recoverin cDNA was synthesized according to the reference sequence (GenBank ID: NM_002903.3) (Integrated DNA Technologies Inc., Coralville, IA, USA) then amplified by polymerase chain reaction (PCR) using the following primers. The forward primer sequence was: gagagaggatccgccaccatggggaacagcaaaagtgg and the reverse primer was gggctcgagtcacaggtcctcctctgagatcagcttctgctcgccggcgttcttcatcttttcct. Underlines indicate BamHI site and XhoI site, respectively. The reverse primer contains a myc tag sequence for protein detection. The PCR was performed using Prime STAR GXL (Takara Bio Inc. Kusatsu, Japan) according to the manufacture’s protocol, the PCR product was then cloned into pMXs-puro vector (Cosmo Bio, Tokyo, Japan) and the sequence was confirmed. For the stable expression of recoverin, A549 cells were transduced retrovirally as described previously (PMID: 10871756), and selected by puromycin (FUJIFILM Wako chemical corp., Hiratsuka, Japan) at a concentration of 1µg/mL. Protein expression was then confirmed by Western blot using anti-myc antibody (A549 Rec). Empty vector transformed A549 cells (A549 MOCK) was used for a negative control. A549 Rec cells and A549 MOCK cells were 2D- and 3D-cultured by methods described in previous reports using 3T3-L1 preadipocytes and human orbital fibroblasts [27,28,49,50,51,52]. Briefly, these cells were each cultured in 2D culture dishes at 37 °C in HG-DMEM culture medium supplemented with 8 mg/L d-biotin, 4 mg/L calcium pantothenate, 100 U/mL penicillin, 100 μg/mL streptomycin (b.p. HG-DMEM), 10% CS and methylcellulose (Methocel A4M) until reaching approximately 90% confluence. They were then divided into conventional 2D cultures and 3D spheroid cultures. The 2D-cultured cells were maintained with medium changes daily until Day 6. Alternatively, for the generation of the 3D spheroids, after washing with a phosphate-buffered saline (PBS), the cells were detached by treatment with 0.25% Trypsin/EDTA, resuspended in the culture medium, and 28 μL medium containing approximately 20,000 cells were added into each well of the drop culture plate (# HDP1385, Sigma-Aldrich) (3D/Day 0) as described previously [27,50]. Thereafter, half of the culture medium was replaced with fresh medium in each well daily until Day 6 [28,51,52]. At Day 7, both 2D- and 3D-cultured cells were collected and further processed for use in the analyses described below.

### 4.2. Measurement of Mitochondrial and Glycolytic Functions of 2D-Cultured A549 Cells

Oxygen consumption rate (OCR) and extracellular acidification rate (ECAR) of 2D-cultured A549 Rec and A549 MOCK as above were simultaneously measured using Seahorse XFe96 analyzer (Agilent Technologies, Santa Clara, CA, USA) according to the manufacturer’s instructions. Briefly, the number of 20 × 10^3^ 2D-cultured A549 cells were placed in a well of a XFe96 Cell Culture Microplate (Agilent Technologies, #103794-100). Following centrifugation of the plate at 1000× *g* for 10 min, the culture medium was replaced with 180 μL of Seahorse XF DMEM assay medium (pH 7.4, Agilent Technologies, #103575-100) supplemented with 5.5 mM glucose, 2.0 mM glutamine, and 1.0 mM sodium pyruvate. The assay plates were incubated in CO_2_-free incubator at 37 °C for 1 hour prior to the measurement. OCR and ECAR were measured by the Seahorse XFe96 Bioanalyzer under protocols of 3 min mix and 3 min measure protocols at the baseline and following injections of oligomycin (final concentration: 2.0 μM), carbonyl cyanide p-trifluoromethoxyphenylhydrazone (FCCP, final concentration: 5.0 μM), a mixture of rotenone/antimycin A (final concentration: 1.0 μM), and 2-deoxyglucose (2-DG, final concentration: 10 mM). The OCR and ECAR values were normalized to the amount of protein per well.

Metabolic parameters using OCR and ECAR were calculated using the following formulas: Basal OCR = OCR at baseline (OCR_basal_) − OCR under rotenone/antimycin A (OCR_r/a_); ATP-linked Respiration = OCR_basal_ α − OCR under oligomycin (OCR_oligo_); Proton Leak = OCR_oligo_ − OCR_r/a_; Maximal Respiration = OCR under FCCP (OCR_FCCP_) − OCR_r/a_; Mitochondrial Spare Reserve Capacity = OCR_FCCP_ − OCR_basal_; Non-mitochondrial Respiration = OCR_r/a_; Basal ECAR = ECAR at baseline (ECAR_basal_) − ECAR at the last measurement under 2-DG (ECAR_2-DG_); Glycolytic Capacity = ECAR under oligomycin (ECAR_oligo_) − ECAR_2-DG_; Glycolytic Reserve = ECAR_oligo_ − ECAR_basal_; Non-glycolytic acidification = ECAR_2-DG_.

The conversion of ECAR into the proton efflux rate (PER) was performed to calculate the rate of glycolytic ATP production using the Seahorse XF Wave software. To evaluate the roles of mitochondrial respiration and glycolysis in ATP production, the following formulas were used: Rate of ATP production during oxidative phosphorylation (mitoATP production rate) = ATP-linked Respiration × 2 × P/O (pmol ATP/pmol O = 2.75); mitochondria-derived acidification (mitoPER) = OCR_basal_ × CO_2_ contribution factor (0.61); Rate of ATP production during glycolysis (glycoATP production rate) = glycolytic proton efflux rate (glycoPER)—mitoPER.

### 4.3. Phase Contrast Microscopy of the 3D Spheroids

The 3D spheroid morphology was observed with a phase contrast microscope (Nikon ECLIPSE TS2; Tokyo, Japan) and a micro-monitoring camera equipped with a microsqueezer (MicroSquisher, CellScale, Waterloo, ON, Canada) as described previously [28,51,52]. For the measurement of the sizes of individual 3D spheroids, the largest cross-sectional area (CSA) of the image was measured and analyzed by an Image-J software version 1.51n (National Institutes of Health, Bethesda, MD, USA).

### 4.4. RNA Sequencing, Gene Function and Analysis of Pathways

Total RNA was isolated from 2D confluent cells of A549 Rec and A549 Mock in a 150 mm dish as described above (n = 3) using a RNeasy mini kit (Qiagen, Valencia, CA, USA) according to the manufacturer’s instructions. RNA content and quality were measured using NanoPhotometer^®^ P330 (IMPLEN, Los Angeles, CA, USA) and the Agilent 2100 Bioanalyzer (Agilent Technologies, Rue Galvani, Massy, France), respectively. Since the RNA quality was suitable for RNA sequencing, and quantitative real-time PCR, the samples with an RNA integrity number (RIN) > 8.5 were confirmed in advance. The total RNA was depleted of ribosomal RNA using NEBNext^®^ Poly(A) mRNA Magnetic Isolation Module (Cat. # E7490, New England BioLabs, Ipswich, MA, USA). The rRNA-depleted RNA was processed according to the manufacturer’s protocol to convert to cDNA using a TruSeq RNA Sample Preparation Kit (Illumina, San Diego, CA, USA) and final sequence-ready libraries with the NEBNext Ultra II RNA library prep kit (Cat. #E7760, New England BioLabs). Their quality and quantity were subsequently determined using an Agilent 2100 Bioanalyzer and KAPA Library Quantification Kit (KAPA Biosystems, Wilmington, MA, USA), respectively. Thereafter, they were subjected to NovaSeq 6000 and GenoLab M sequencing in the PE150 mode. Sequence data were filtered by removal of the adapter sequence, ambiguous nucleotides, and low-quality sequences using softwares; FastQC (version 0.11.7) as quality control by an Agilent 2100 Bioanalyzer (Agilent, CA, USA) and Trimmomatic (version 0.38). These clean reads were mapped to the reference genome sequence (GRCh38) using HISAT2 tools software [53] (Appendix A). The read counts for each respective gene and statistical analysis of the differentially expressed genes were calculated by featureCounts (version 1.6.3) and DESeq2 (version 1.24.0), respectively. Statistical significance was determined by an empirical analysis, and genes with fold-change ≥ 2.0 and FDR-adjusted *p*-value < 0.05 and *q* < 0.08 were assigned as differentially expressed genes (DEG). 

To estimate gene function, gene ontology (GO) enrichment analysis [54] as well as ingenuity pathway analysis (IPA, Qiagen, https://www.qiagenbioinformatics.com/products/ingenuity-pathway-analysis, accessed on 19 August 2022) [55] were performed as described recently [49]. Briefly, during the GO enrichment analysis, molecular function, biological processes, cellular components and others with a *p* value of less than 0.05 were considered as significantly enriched by differential expressed genes based upon the Kyoto Encyclopedia of Genes and Genomes (KEGG), a database resource that contains information on high-level functions and the effects of a biological system accessed on 19 August 2022 (http://www.genome.jp/kegg/).

To predict possible upstream transcriptional regulators, DEGs were interpreted using the upstream regulator function of the ingenuity pathway analysis accessed on 19 August 2022 (IPA, Qiagen, https://www.qiagenbioinformatics.com/products/ingenuity-pathway-analysis) [55]. Significance of the biofunctions and the canonical pathways was evaluated by Fisher’s exact test *p*-value. Biofunctions were categorized in: Disease and Disorders; Molecular and Cellular Functions; and Physiological System Development and Function. Alternatively, the canonical pathways were categorized in Metabolic Pathways and Signaling Pathways. Canonical pathways can also be ordered by the ratio value (number of molecules in a given pathway that meet cut criteria, divided by the total number of molecules that make up that pathway) [55].

### 4.5. Other Analytical Methods

Cytotoxicity assays using 2D cultures A549 Rec and A549 MOCK against several anti-tumor drugs including carboplatin, pemetrexed and docetaxel were determined using a Cell Counting Kit-8 (Dojindo, Tokyo, Japan). All statistical analyses were performed using the Graph Pad Prism 8 (GraphPad Software, San Diego, CA, USA). The statistical difference between groups were determined using a Student’s *t*-test for two group comparison or two-way ANOVA followed by a Tukey’s multiple comparison test was performed. Data are expressed as the arithmetic mean ± the standard error of the mean (SEM).

## Figures and Tables

**Figure 1 ijms-24-00771-f001:**
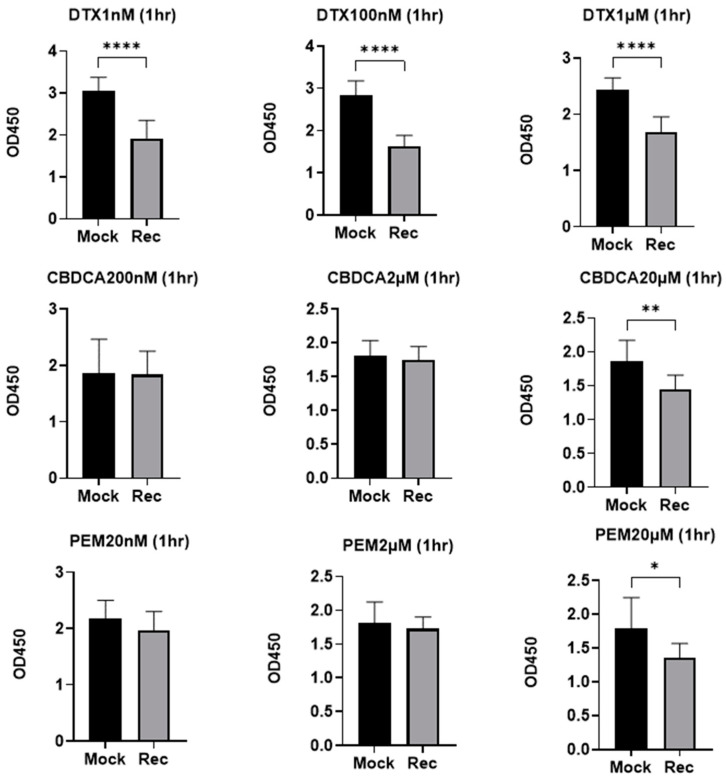
Cytotoxicity caused by several anti-tumor agents toward A549 MOCK and A549 Rec. To estimate the cytotoxic effects of several anti-tumor agents at different concentrations, including decetaxel (DTX; 1 nM, 100 nM and 1 μM), carboplatin (CBDCA; 200 nM, 2 μM and 20 μM), and pemetrexed (PEM; 20 nM, 2 μM and 20 μM) on 2D-cultured A549 MOCK and A549 Rec, the survival of the Cells was evaluated by means of a Cell Counting Kit-8, (Dojindo, Tokyo, Japan) according to the manufacturer’s protocol. Briefly, after culturing 2D cells (5000 cells/well) for 18 h in the absence or presence of each anti-tumor agent, they were incubated with 10 μL of reactive solution for 1 h. The absorbance of each cell at 450 nm was measured using a microplate reader (multimode plate reader EnSpire^®^, PerkinElmer, Waltham, MA, USA) and plotted (n = 4). Data are expressed as the mean ± the standard error of the mean (SEM). * *p* < 0.05, ** *p* < 0.01, **** *p* < 0.001 (Student’s *t*-test).

**Figure 2 ijms-24-00771-f002:**
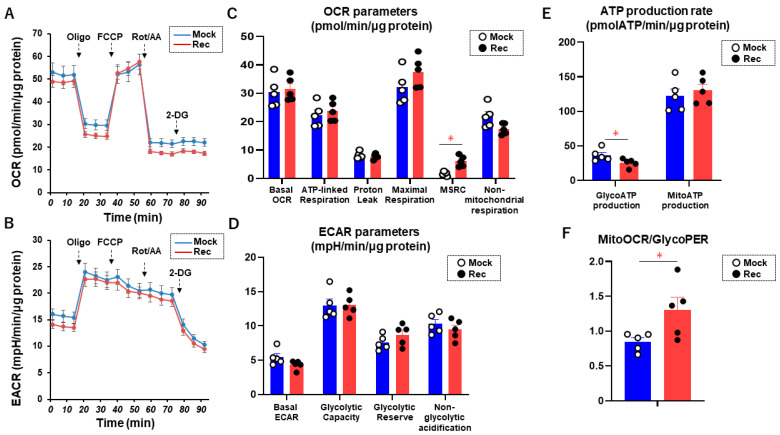
Measurement of mitochondrial and glycolytic functions of A549 MOCK and A549 Rec. The 2D-cultured A549 MOCK and A549 Rec cells were subjected to mitochondrial and glycolytic function analyses using a Seahorse XFe96 analyzer. Measurements of oxygen consumption rate (OCR, panel (**A**)) and extracellular acidification rate (ECAR, panel (**B**)) before drug injections (at baseline) and after subsequent supplementation with oligomycin (complex V inhibitor), FCCP (a protonphore), and rotenone/antimycin A (complex I/III inhibitors) and 2-DG (hexokinase inhibitor) are shown. Key parameters in mitochondrial respiration and glycolysis are shown in panel (**C**,**D**), respectively. MitoATP production rate and GlycoATP production rate are shown in panel (**E**). Basal mitoOCR/glycoPER ratio is shown in panel (**F**). Experiments were performed using fresh preparations (n = 5). Data are presented as the mean ± the standard error of the mean (SEM). * *p* < 0.05 (Student’s *t*-test).

**Figure 3 ijms-24-00771-f003:**
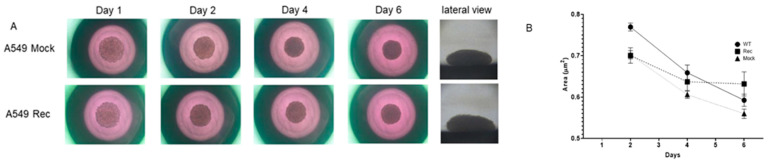
Comparisons of the mean sizes of 3D spheroids of WT, A549 MCOK and A549 Rec. In panel (**A**), representative phase contrast microscopy images of A549 MOCK and A549 Rec are shown (horizontal views at Day 1, 2, 4 and 6, and lateral views at Day 6, scale bar; 100 μm). The mean sizes of the 3D spheroids of WT, A549 MOCK and A549 Rec were measured at Day 2, Day 4 and Day 6, and plotted in panel (**B**). Experiments were repeated in triplicate using fresh preparations (n = 10 spheroids each). All data are expressed as the mean ± the standard error of the mean (SEM). Statistical significance was evaluated by ANOVA followed by a Turkey’s multiple comparison test.

**Figure 4 ijms-24-00771-f004:**
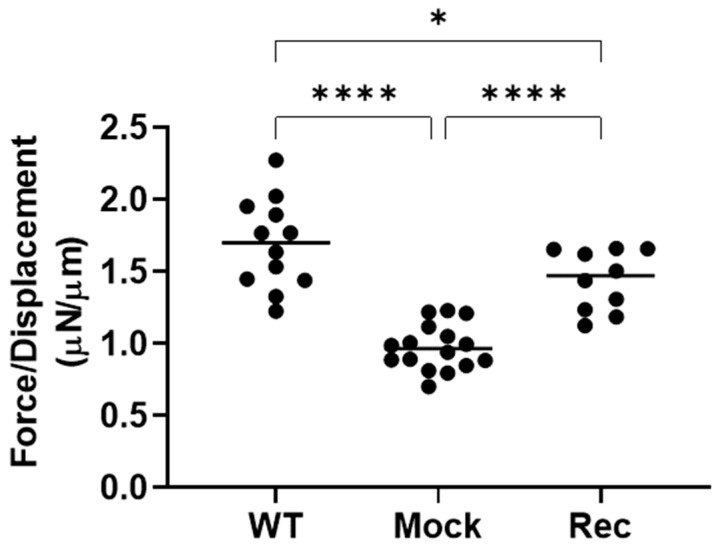
The stiffness of the 3D spheroids of WT, A549 MOCK and A549 Rec were measured at Day 6 using a micro-squeezer, as described in the Methods section. The force (μN) required to compress a single spheroid to the semidiameter (μN/μm) within 20 s are plotted. Experiments were repeated in triplicate using fresh preparations (n = 10 spheroids each). All data are expressed as the mean ± the standard error of the mean (SEM). Statistical significance was evaluated by ANOVA followed by a Turkey’s multiple comparison test (* *p* < 0.05, **** *p* < 0.001).

**Figure 5 ijms-24-00771-f005:**
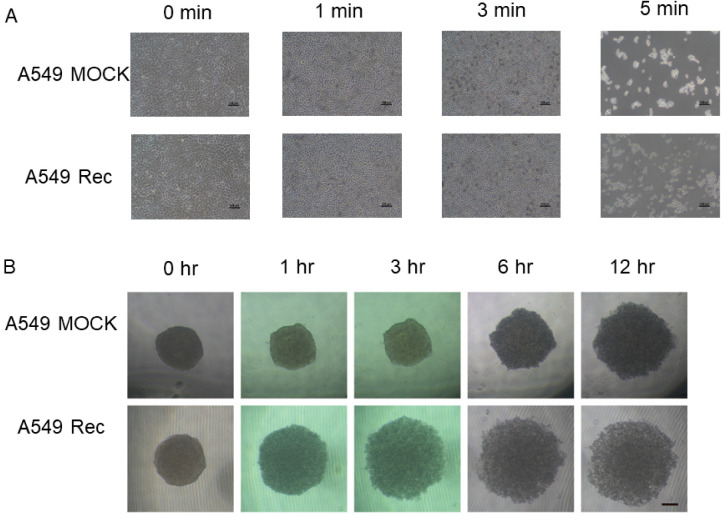
Trypsin digestion of 2D- or 3D-cultured A549 MCOK and A549 Rec. The 2D- or 3D-cultured A549 MOCK and A549 Rec were treated with 0.025% trypsin for 5 min or 12 h, respectively. Representative phase contrast microscopy images of 2D cells and 3D spheroids are shown in panels (**A**,**B**), respectively (scale bar; 100 μm). Experiments were repeated in triplicate using fresh preparations (2D; n = 5, 3D; n = 10 spheroids each).

**Figure 6 ijms-24-00771-f006:**
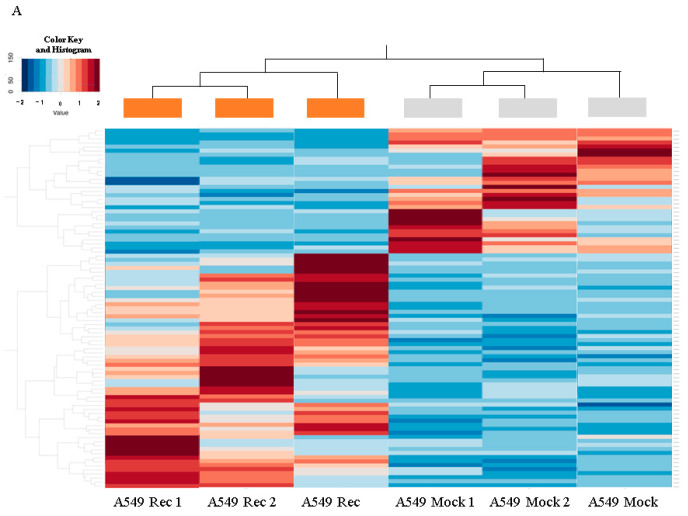
Differently expressed genes (DEGs) between A549 MOCK and A549 Rec cells. In total, 88 genes were expressed differently (fold-change ≥ 2.0 and FDR adjusted *p*-Value < 0.05 and *q* < 0.08) between A549 Mock and A549 Rec cells are shown by a hierarchical clustering heatmap (panel (**A**), A549 Mock; gray, A549 Rec; orange), M–A plots (panel (**B**)) and volcano plots (panel (**C**)). M–A plot represents the relationship between the mean expression values [log (baseMean); *x*-axis] and the magnitude of change in gene expression (log2 of fold-change; *y*-axis), and volcano plot represents the relationship between the magnitude of gene expression change (log2 of fold-change; *x*-axis) and statistical significance of this change [−log10 of false discovery rate (FDR); *y*-axis].

## Data Availability

The data that support the findings of this study are available from the corresponding author upon reasonable request.

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
