# Peer review of "G-Protein-Coupled Receptors Mediate Modulations of Cell Viability and Drug Sensitivity by Aberrantly Expressed Recoverin 3 within A549 Cells"

_ijms, 2023, doi:10.3390/ijms24010771_

Round 1

Reviewer 1 Report

The current manuscript investigates the role of aberrantly expressed Recoverin in the human A549 lung adenocarcenoma cell line. Recoverin is the major investigated autoantigen in cancer-associated retinopathy, and has been shown to be aberrantly expressed in many cancer cell lines. The current manuscript shows that aberrant high expression of Recoverin in A549 cells leads to increased sensitivity to different anti-tumor agents, reduced ATP production during glycolysis, increased Mitochondrial Spare Reserve Capacity, increased size of spheroids, increased sensitivity of spheroids to Trypsin, and an upregulation for GPCR, CREB and Gia signaling. The authors conclude that Recoverin-expressing A549 cells shift from glycolysis to oxidative phosphorylation and change cellular functions and structures due to changed GPCR, CREB and Gia signaling.

The study is well performed and reasonably written. However, there are several drawbacks:

First, only a single cell line was analyzed, that very highly expresses Recoverin after transfection (as seen in the RNAseq) – it would be essential to show, whether the phenotype is also seen in other cell lines (either non-Recoverin-expressing cancer cell line transfected with Recoverin, or Recoverin-expressing cancer cell line with silenced Recoverin) and if the phenotype is also seen if Recoverin is expressed at lower levels in A549 cells.

Secondly, I missed the connection between the Seahorse data and the RNAseq/pathway analysis.

Minor comments:

-        Please refer to RNA sequence analysis as RNA sequencing analysis

-        Please also show the results of the GO enrichment analysis/ Ingenuity Pathway analysis (FDR, p-values, enrichment scores, …) and provide some more methodological insights

-        Please refer to 293 cells as HEK293 cells

-        Please correct several structural/spelling errors.

Author Response

Dear Editor,

Thank you very much for the constructive comments concerning our manuscript, " G-protein coupled receptors mediated modulations of cell viability and drug sensitivity by the aberrant expressed recoverin within A549 cells”. We examined the Reviewer's comments carefully and prepared a revised version of our paper that takes these comments into account for resubmission. Therefore, we will greatly appreciate it if you will consider our revised paper for possible publication in IJMS. The changes are listed below.

Reviewer 1

The current manuscript investigates the role of aberrantly expressed Recoverin in the human A549 lung adenocarcenoma cell line. Recoverin is the major investigated autoantigen in cancer-associated retinopathy, and has been shown to be aberrantly expressed in many cancer cell lines. The current manuscript shows that aberrant high expression of Recoverin in A549 cells leads to increased sensitivity to different anti-tumor agents, reduced ATP production during glycolysis, increased Mitochondrial Spare Reserve Capacity, increased size of spheroids, increased sensitivity of spheroids to Trypsin, and an upregulation for GPCR, CREB and Gia signaling. The authors conclude that Recoverin-expressing A549 cells shift from glycolysis to oxidative phosphorylation and change cellular functions and structures due to changed GPCR, CREB and Gia signaling.

The study is well performed and reasonably written. However, there are several drawbacks:

  1. First, only a single cell line was analyzed, that very highly expresses Recoverin after transfection (as seen in the RNAseq) – it would be essential to show, whether the phenotype is also seen in other cell lines (either non-Recoverin-expressing cancer cell line transfected with Recoverin, or Recoverin-expressing cancer cell line with silenced Recoverin) and if the phenotype is also seen if Recoverin is expressed at lower levels in A549 cells.

Answer; Thank you for this comment. I agree that our experimentally prepared A549 Rec cell is overexpress system and naturally Rec expressed cancer cell lines should be much lower expression levels of Rec. However, our previous studies demonstrated that immunostaining of Rec of the surgically obtained cancerous tissues were higher within earlier clinical cancerous stages, and this may rationally support current data of the higher sensitivities of A549 Rec against anti-tumor drugs (Fig. 1). Therefore, this information is included within the study limitation within the Discussion; “However, current hypotheses are speculative at present because of following study limitations that would need to be investigated for overcoming them; 1) our current experimental system used very highly expresses Rec after transfection. Therefore, it should be better to understand the pathophysiological aspects of the aberrantly expressed Rec within cancerous cells to use those with lower Rec expressions or naturally and aberrantly Rec expressions. Although we do not know the phenotypes of such lower Rec expressed tumor cells, our previous studies that immunostaining of Rec of the surgically obtained cancerous tissues were higher within earlier clinical cancerous stages [22,49] rationally support current data of the higher sensitivities of A549 Rec against anti-tumor drugs (Fig. 1). 2) The relationship between the RNAseq/pathway analysis and the difference of cellular metabolic states by the Seahorse Bioanalyzer measurements between A549 MOCK and A549 Rec is not fully understood yet. However, some mitochondria related genes were identified within the DEGs (Table 1). In addition, significant up-regulated retinal Rec expression was found in knockout mice of Ant1 (Adenine nucleotide translocator 1) expressed within the inner mitochondrial membrane [50,51], in which oxidative phosphorylation (OXPHOS) was decreased as compared with WT [52,53]. These observations may support our current results. 3) As shown in Figs 4 and 5, the stiffness of the A549 MOCK 3D spheroids is lower but their trypsin resistance is higher as compared with A549 Rec. Although these two physical aspects may not simply be understood, we speculated that much complicated underlying mechanisms including plasticity as well as elasticity may be involved within the current stiffness measurement of the 3D spheroids. In fact, upon administration of trypsin, we observed earlier swelling of the 3D A549 Rec spheroid as compared with 3D A549 MOCK, but both 3D spheroids did not come apart until 12 hours. In addition, genes related to several integral and anchoring components of membrane were identified among DEGs. Therefore, additional functional and morphological investigations by modulating the currently identified genes as well as several possible related biological pathways will be required as our next project.”

  1. Secondly, I missed the connection between the Seahorse data and the RNAseq/pathway analysis.

Answer; Thank you for this comment. In terms of this issue, the connection between the Seahorse data and the RNAseq/pathway analysis was not fully understood yet. However, some mitochondria related genes were identified within the DEGs (Table 1), and interesting previous report related to the recoverin expression and mitochondrial function was demonstrated. Therefore, this information is included within the study limitation within the Discussion; “However, current hypotheses are speculative at present because of following study limitations that would need to be investigated for overcoming them; 1) our current experimental system used very highly expresses Rec after transfection. Therefore, it should be better to understand the pathophysiological aspects of the aberrantly expressed Rec within cancerous cells to use those with lower Rec expressions or naturally and aberrantly Rec expressions. Although we do not know the phenotypes of such lower Rec expressed tumor cells, our previous studies that immunostaining of Rec of the surgically obtained cancerous tissues were higher within earlier clinical cancerous stages [22,49] rationally support current data of the higher sensitivities of A549 Rec against anti-tumor drugs (Fig. 1). 2) The relationship between the RNAseq/pathway analysis and the difference of cellular metabolic states by the Seahorse Bioanalyzer measurements between A549 MOCK and A549 Rec is not fully understood yet. However, some mitochondria related genes were identified within the DEGs (Table 1). In addition, significant up-regulated retinal Rec expression was found in knockout mice of Ant1 (Adenine nucleotide translocator 1) expressed within the inner mitochondrial membrane [50,51], in which oxidative phosphorylation (OXPHOS) was decreased as compared with WT [52,53]. These observations may support our current results. 3) As shown in Figs 4 and 5, the stiffness of the A549 MOCK 3D spheroids is lower but their trypsin resistance is higher as compared with A549 Rec. Although these two physical aspects may not simply be understood, we speculated that much complicated underlying mechanisms including plasticity as well as elasticity may be involved within the current stiffness measurement of the 3D spheroids. In fact, upon administration of trypsin, we observed earlier swelling of the 3D A549 Rec spheroid as compared with 3D A549 MOCK, but both 3D spheroids did not come apart until 12 hours. In addition, genes related to several integral and anchoring components of membrane were identified among DEGs. Therefore, additional functional and morphological investigations by modulating the currently identified genes as well as several possible related biological pathways will be required as our next project.”

Minor comments:

  1. Please refer to RNA sequence analysis as RNA sequencing analysis

Answer; Thank you for this comment. As pointed out, “RNA sequence analysis” was changed to “RNA sequencing analysis”

  1. Please also show the results of the GO enrichment analysis/ Ingenuity Pathway analysis (FDR, p-values, enrichment scores, …) and provide some more methodological insights

Answer; Thank you for this comment. As suggested, those information were included in the method; “Briefly, during the GO enrichment analysis, molecular function, biological processes, cellular components and others with a P value less than 0.05 were considered to be significantly enriched by differential expressed genes based upon the Kyoto Encyclopedia of Genes and Genomes (KEGG), a database resource for understanding high-level functions and the effects of the biological system (http://www.genome.jp/kegg/).

To predict possible upstream transcriptional regulators, DEGs were interpreted using the upstream regulator function of the ingenuity pathway analysis (IPA, Qiagen, https://www.qiagenbioinformatics.com/products/ingenuity-pathway-analysis). As predicted function, pathway or upstream regulator (activation or deactivation), the activation z-score algorithm (the number of standard deviations in the data lies above or below the mean. A z-score ≥2 was considered to be significantly increased whereas a z-score ≤ −2 was considered significantly decreased [18]) was used.”

  1. Please refer to 293 cells as HEK293 cells

Answer; Thank you for this comment. As pointed out, “293 cells” was changed to “HEK293 cells”.

  1. Please correct several structural/spelling errors.

Answer; Thank you for this comment. As suggested, the quality of English was edited by native speaking research scientific professor, Milton S Feather; A letter of confirmation (in PDF format) from him is attached.

Reviewer 2

The paper by Ichioka and colleagues uses transfection studies on A549 cells to study the role of the Recoverin protein. The authors use metabolic analysis, 2d and 3d culture conditions and drug sensitivity assays, together with RNA seq to try to identify the impact of Rec expression on cell behaviour. The data collected are incomplete and there is no explanation for many of the observations. The RNAseq doesn’t add to the paper at all because none of the genes that are altered are able to explain any of the observations. Indeed, the whole paper seems like a loosely connected series of observations. I found the discussion very strange. It seems to be written to fit the authors’ preconception about the role of recoverin, rather than having any solid connection to the data presented.

Major

  1. Figure 1 cytotoxicity is with 3 concentrations, rather than a full dose-response curve. Can the authors comment on why they have carried out the work like this as it prevents them from reaching a quantitative conclusion regarding the degree of sensitivity. I think this type of full dose-response curve is absolutely essential to argue that Rec expression is connected to increased drug sensitivity.

Answer; Thank you for this comment. As suggested, Fig.1 was changed to dose-response curve manner.

  1. In figure 2 the OCR and EACR graphs for MOCK and Rec cells are close to superimposable and the subsequent metabolic shift is rather marginal. Claims about metabolic shift are rather over-stated in my opinion.

Answer; Thank you for this comment. I agree with that may be overstatement. Therefore, those were changed to “metabolic changes between glycolysis and oxidative phosphorylation (OXPHOS)”.

  1. Figure 4-5 the stiffness and trypsin sensitivity are presented. Whilst the stiffness of the Mock 3D spheroids is lower the trypsin resistance is higher. These two things are hard to reconcile. What explanation do the authors have for this?

Answer; Thank you for this comment. In this issue, we do not know the exact mechanisms causing these phenomena. However, we speculated that as possible underlying mechanisms, plasticity as well as elasticity may be involved within the current stiffness measurement of the 3D spheroids. In fact, upon administration of trypsin, we observed earlier swelling of the 3D A549 Rec spheroid as compared with 3D A549 MOCK, but both 3D spheroids did not come apart until 12 hours. In addition, genes related to several integral and anchoring components of membrane were identified among DEGs. Therefore, this information is included in the study limitations within the Discussion; “However, current hypotheses are speculative at present because of following study limitations that would need to be investigated for overcoming them; 1) our current experimental system used very highly expresses Rec after transfection. Therefore, it should be better to understand the pathophysiological aspects of the aberrantly expressed Rec within cancerous cells to use those with lower Rec expressions or naturally and aberrantly Rec expressions. Although we do not know the phenotypes of such lower Rec expressed tumor cells, our previous studies that immunostaining of Rec of the surgically obtained cancerous tissues were higher within earlier clinical cancerous stages [22,49] rationally support current data of the higher sensitivities of A549 Rec against anti-tumor drugs (Fig. 1). 2)  The relationship between the RNAseq/pathway analysis and the difference of cellular metabolic states by the Seahorse Bioanalyzer measurements between A549 MOCK and A549 Rec is not fully understood yet. However, some mitochondria related genes were identified within the DEGs (Table 1). In addition, significant up-regulated retinal Rec expression was found in knockout mice of Ant1 (Adenine nucleotide translocator 1) expressed within the inner mitochondrial membrane [50,51], in which oxidative phosphorylation (OXPHOS) was decreased as compared with WT [52,53]. These observations may support our current results. 3) As shown in Figs 4 and 5, the stiffness of the A549 MOCK 3D spheroids is lower but their trypsin resistance is higher as compared with A549 Rec. Although these two physical aspects may not simply be understood, we speculated that much complicated underlying mechanisms including plasticity as well as elasticity may be involved within the current stiffness measurement of the 3D spheroids. In fact, upon administration of trypsin, we observed earlier swelling of the 3D A549 Rec spheroid as compared with 3D A549 MOCK, but both 3D spheroids did not come apart until 12 hours. In addition, genes related to several integral and anchoring components of membrane were identified among DEGs. Therefore, additional functional and morphological investigations by modulating the currently identified genes as well as several possible related biological pathways will be required as our next project.”

  1. Table 1 and supplementary figures 2-4 (the RNAseq) data do not give me any confidence in the conclusions reached. None of the genes listed in Table 1 are visible in the 3 supplementary figures. There is no narrative or quantitative data that indicates the regulation of the three pathways is affected. How many genes in each pathway are DEGs in the dataset? What p-values were obtained for the three pathways? The RNAseq doesn’t provide any sort of explanation for any of the other observations presented.

Answer; Thank you for this comment. As suggested, I agree with that Table 1 was insufficient to convince supplemental figures 2-4. Therefore, Table 1 was replaced to new Table 1 including significant up-regulated and down-regulated DEGs (p> 0.05) which contains several genes related to supplemental figures 2-4.

  1. The discussion is a very strange piece of writing that seems to suggest that more experiments are needed to connect the RNAseq with the observations in the paper. I agree with this basic conclusion, but find the other conclusions reached in the discussion to be unsupported. There is no justification for talking at length about STAT signalling or ANTs or caveolins because none of these genes has been identified in RNAseq – they just happen to fit the authors’ narrative about the role of recoverin.

Answer; Thank you for this comment. I agree that STAT signalling or ANTs or caveolins related staffs may be overstatements, therefore those were removed or minimized.

 Minor

  1. Supplemental figure 1 includes the word “Turkey” rather than “Tukey”

Answer; Thank you for this comment. As pointed out, “Tukey” was changed to “Tukey”.

  1. Figure 1. There is no reason for ANOVA when there are only 2 comparison groups. The correct test through out this figure is

Answer; Thank you for this comment. As pointed out, statistical analysis was performed by a Student’s t-test.

  1. Figure 3 Panel C needs removing – it shows the same data as panel B

Answer; Thank you for this comment. As suggested, the corresponding panel C was removed

Reviewer 2 Report

The paper by Ichioka and colleagues uses transfection studies on A549 cells to study the role of the Recoverin protein. The authors use metabolic analysis, 2d and 3d culture conditions and drug sensitivity assays, together with RNA seq to try to identify the impact of Rec expression on cell behaviour. The data collected are incomplete and there is no explanation for many of the observations. The RNAseq doesn’t add to the paper at all because none of the genes that are altered are able to explain any of the observations. Indeed, the whole paper seems like a loosely connected series of observations. I found the discussion very strange. It seems to be written to fit the authors’ preconception about the role of recoverin, rather than having any solid connection to the data presented.

Major

Figure 1 cytotoxicity is with 3 concentrations, rather than a full dose-response curve. Can the authors comment on why they have carried out the work like this as it prevents them from reaching a quantitative conclusion regarding the degree of sensitivity. I think this type of full dose-response curve is absolutely essential to argue that Rec expression is connected to increased drug sensitivity.

In figure 2 the OCR and EACR graphs for MOCK and Rec cells are close to superimposable and the subsequent metabolic shift is rather marginal. Claims about metabolic shift are rather over-stated in my opinion.

Figure 4-5 the stiffness and trypsin sensitivity are presented. Whilst the stiffness of the Mock 3D spheroids is lower the trypsin resistance is higher. These two things are hard to reconcile. What explanation do the authors have for this?  

Table 1 and supplementary figures 2-4 (the RNAseq) data do not give me any confidence in the conclusions reached. None of the genes listed in Table 1 are visible in the 3 supplementary figures. There is no narrative or quantitative data that indicates the regulation of the three pathways is affected. How many genes in each pathway are DEGs in the dataset? What p-values were obtained for the three pathways? The RNAseq doesn’t provide any sort of explanation for any of the other observations presented.

 The discussion is a very strange piece of writing that seems to suggest that more experiments are needed to connect the RNAseq with the observations in the paper. I agree with this basic conclusion, but find the other conclusions reached in the discussion to be unsupported. There is no justification for talking at length about STAT signalling or ANTs or caveolins because none of these genes has been identified in RNAseq – they just happen to fit the authors’ narrative about the role of recoverin.

Minor

Supplemental figure 1 includes the word “Turkey” rather than “Tukey”

Figure 1. There is no reason for ANOVA when there are only 2 comparison groups. The correct test through out this figure is a Student’s t-test.

Figure 3 Panel C needs removing – it shows the same data as panel B

Author Response

(The authors gave the same response as above.)

Round 2

Reviewer 1 Report

Although the discussion has been clearly improved, I see the need for additional experiments in at least one more naturally Rec expressing cancer cell line in order to publish this manuscript in IJMS.

Minor:

Please format the table in a readable was. Reduce the information of Cellular Component Term, Molecular Function Term and Biological Process Term.

Can you comment on the fact that some G-protein coupled receptor-related genes are up-regulated while some are down-regulated in Rec-aberrantly expressing cells?

Author Response

Dear Editor,

Thank you very much for the constructive comments concerning our manuscript, " G-protein coupled receptors mediated modulations of cell viability and drug sensitivity by the aberrant expressed recoverin within A549 cells”. We examined the Reviewer's comments carefully and prepared a revised version of our paper that takes these comments into account for resubmission. Therefore, we will greatly appreciate it if you will consider our revised paper for possible publication in IJMS. The changes are listed below.

Reviewer 1

  1. Although the discussion has been clearly improved, I see the need for additional experiments in at least one more naturally Rec expressing cancer cell line in order to publish this manuscript in IJMS.

Answer; Thank you so much for this comment. We agree that it would be great to add additional information related to the characterization of the naturally Rec expressing cancer cell line. However, in our previous study, we identified several Rec positive cell lines by qPCR analysis, but we have no information concerning the level of Rec expression within these cell lines. We assumed that such Rec expression levels should be different and not the same among cell lines because Rec is exclusively a retina specific molecule. Therefore, even if several naturally Rec positive cell lines were to be identified, their biological feature may be different. Nevertheless, we found a very interesting study by Yamaji et al (Int J Cancer, 1996 Mar 1;65(5):671-6.). In this study, they established a naturally Rec expressing small-cell-lung-carcinoma (SCLC) cell line, designated MN-1112, from a patient with SCLC who showed the CAR syndrome. The characterization of the biological aspects indicated that the morphologic and immunocytochemical features, and efficacies of cell growth, production of tumor makers, and carcinogenesis toward nude mice of MN-1112 cells were quite similar to those of the classic type of SCLC cell lines. Therefore, this information is included within the 2nd paragraph of Discussion; “However, current hypotheses remain speculative at present because of the following study limitations that would need to be investigated for them to be overcome; 1) our current experimental system expressed high levels of Rec after transfection. Therefore, in order to understand the pathophysiological aspects of the aberrantly expressed Rec within cancerous cells, it would be better to use cells with cells that express lower levels of Rec or that naturally and aberrantly express Rec. Although we have very limited knowledge concerning the phenotypes of these types of tumor cells that express lower levels of Rec, Yamaji et al. established a small-cell-lung-carcinoma (SCLC) cell line, designated MN-1112, that naturally express Rec from a patient with SCLC who showed CAR syndrome. In the characterization of the biological aspects of these cells, the morphologic and immunocytochemical features, and efficacies of cell growth, production of tumor makers, and carcinogenesis toward nude mice of MN-1112 cells were examined and were found to be quite similar to those of the classic type of SCLC cell lines [20]. However, in our previous studies that involved the immunostaining of Rec from surgically obtained cancerous tissues the expression was higher in the case of earlier clinical cancerous stages [22,52] which rationally support the current data showing higher sensitivities of A549 Rec against anti-tumor drugs (Fig. 1). 2)  The relationship between the analysis of the RNAseq/pathway and the difference in cellular metabolic states based on Seahorse Bioanalyzer measurements between A549 MOCK and A549 Rec is not fully understood at this time. However, some mitochondria related genes were identified within the DEGs (Table 1). In addition, in Ant1 (Adenine nucleotide translocator 1) knockout mice, a significantly up-regulated retinal Rec expression was found to be expressed within the inner mitochondrial membrane [53,54], in which oxidative phosphorylation (OXPHOS) was decreased as compared with WT [55,56]. These observations are consistent with our current results. 3) As shown in Figs 4 and 5, the stiffness of the A549 MOCK 3D spheroids was lower but their trypsin resistance was higher as compared with A549 Rec. Although these two physical aspects may not be completely understood, we speculate that more complicated underlying mechanisms including plasticity as well as elasticity may be involved in the case of the stiffness measurements of these 3D spheroids. In fact, upon the administration of trypsin, we observed the earlier swelling of the 3D A549 Rec spheroids as compared with 3D A549 MOCK, but both 3D spheroids remained intact for periods of up to 12 hours. In addition, genes related to several integral and anchoring components of membranes were identified among the DEGs. Therefore, additional functional and morphological investigations that involve modulating the currently identified genes as well as several possible related biological pathways will be required as our next project.”.

Minor:

  1. Please format the table in a readable was. Reduce the information of Cellular Component Term, Molecular Function Term and Biological Process Term.
  2. Can you comment on the fact that some G-protein coupled receptor-related genes are up-regulated while some are down-regulated in Rec-aberrantly expressing cells?

Answers for #2 and #3; Thank you for these constructive comments, and we apologized for misunderstanding how I should properly answer these questions. I would like to honestly show several DEGs related to G-protein coupled receptor signaling, CREB signaling and Gia mediated signaling, and therefore I prepared such a very complex Table 1. However, as pointed out, we agree that this Table 1 is difficult to read. We do, however, I think that this information is still important. In addition, information related to the methods and results in terms of canonical pathway by IPA analysis was insufficiently described. Therefore, this Table 1 is included within the supplemental material, and more information related to the obtained three significant canonical pathway by IPA analysis with -log(p-value) were included in the method; “To predict possible upstream transcriptional regulators, DEGs were interpreted using the upstream regulator function of the ingenuity pathway analysis (IPA, Qiagen, https://www.qiagenbioinformatics.com/products/ingenuity-pathway-analysis) [33]. The significance of the biofunctions and the canonical pathways were evaluated by the Fisher Exact test p-value. Biofunctions were categorized as: Disease and Disorders; Molecular and Cellular Functions; and, Physiological System Development and Function. Alternatively, the canonical pathways were categorized as Metabolic Pathways and Signaling Pathways. Canonical pathways can also been ordered by the ratio (the number of molecules in a given pathway that meet cut criteria, divided by the total number of molecules that make up that pathway) [33].“, and last paragraph of Results; “To elucidate the currently unidentified mechanisms responsible for inducing such characteristic features as above between A549 Rec and A549 MOCK, RNA sequence analyses were performed. As shown in MA and volcano plots (Fig. 6 A, B), 32 significantly up-regulated and 50 down-regulated differentially expressed genes (DEGs) were identified for A549 Rec as compared to A549 MOCK with a significance level of <0.05 (FDR) and an absolute fold-change ≥2 was identified (the list of the up-regulated and down-regulated genes is shown in supplemental Table 1). To estimate the possible functional roles of the above DEGs that were detected, we conducted a GO enrichment analysis and an Ingenuity Pathway Analysis (IPA) and (Qiagen, Redwood City, CA). The top three significant canonical pathways, G-protein coupled receptor (GPCR) signaling (-log(p-value) = 6.3), CREB signaling (-log(p-value) = 5.8), Gia mediated signaling (-log(p-value) = 5.2) were identified based upon the detected DEGs (supplemental Figs. 2-4). “. 

Reviewer 2 Report

Thank you for the revised version of this manuscript. The manuscript contains several minor changes in formatting and phrasing but doesn't actually address the major concerns raised at initial review:

i) In the revised manuscript the authors have presented the same drug response data a different way. This isn't a full dose-response curve analysis so I don’t think they can make anything other than very tentative conclusions about recoverin expression and drug sensitivity. Indeed for CBDCA and PEM there is no good evidence for recoverin affecting response to drugs

ii) the data are all still disconnected and observational. The new version includes a revised Table now spreading over 12 pages which still doesn’t address the issue raised which is that the RNAseq data doesn’t provide any explanation of the phenomena observed.  

I considered the initial manuscript to require major changes and this version seems to have had only cosmetic changes. To me, it does not warrant publication. 

Author Response

Dear Editor,

Thank you very much for the constructive comments concerning our manuscript, " G-protein coupled receptors mediated modulations of cell viability and drug sensitivity by the aberrant expressed recoverin within A549 cells”. We examined the Reviewer's comments carefully and prepared a revised version of our paper that takes these comments into account for resubmission. Therefore, we will greatly appreciate it if you will consider our revised paper for possible publication in IJMS. The changes are listed below.

Reviewer 2

Thank you for the revised version of this manuscript. The manuscript contains several minor changes in formatting and phrasing but doesn't actually address the major concerns raised at initial review:

  1. in the revised manuscript the authors have presented the same drug response data a different way. This isn't a full dose-response curve analysis so I don’t think they can make anything other than very tentative conclusions about recoverin expression and drug sensitivity. Indeed for CBDCA and PEM there is no good evidence for recoverin affecting response to drugs.

Answer; Thank you for this comment, and we apologize for misunderstanding how to answer this comment. As pointed out, we agree that the current experimental data do not describe a full dose-response curve. However, this experiment was not new but a re-examination to confirm previous our data in our previous study (Ophthalmic Res. 2010;43(3):139-44.). In that study, we performed a full dose-response curve analysis of several anti-tumor drugs including DTX. In fact, we repeatedly found that Rec induced an increase in their drug sensitivities in the current DTX related data despite the fact that we tested only three concentration points. Therefore, prior to a full dose-response curve analysis, we were interested in obtaining information on Rec induced differences in drug sensitivities using the three points of drug concentrations, especially in terms of CBDCA and PEM that were not evaluated in our previous study. Interestingly, as pointed out, the CBDCA and PEM induced cytotoxic effects were not significantly different between A549 MOCK and A549 Rec, suggesting that aberrantly expressed Rec may influence specific anti-tumor mechanisms rather than all of those mechanisms. Since this pilot study related to CBDCA and PEM did not show any difference between A549 MOCK and A549 Rec, we concluded that an additional full dose-response curve analysis would not be required. Therefore, this information is included in the 1st paragraph of Result; “To study the effects of the aberrant expression of Rec within A549 lung adenoma cells, human recoverin cDNA was transfected and a positive expression was confirmed within A549 Rec, but not A549 WT nor A549 MOCK by a qPCR analysis (supplemental Fig. 1), and these calls were subjected to 2D and 3D cell cultures. Initially, cytotoxicity by several anti-tumor agents, and real time cellular metabolic functions of the 2D cultured cells were evaluated (Fig. 1). Cytotoxicity that had been induced by DTX in the 2D cultured A549 Rec was increased compared to control MOCK cells, as was also observed in our previous studies [22,23]. However in contrast, significant differences in the sensitivities caused by the tumor drugs, CRDCA or PEM, were not observed between A549 MOCK and A549 Rec, suggesting that aberrantly expressed Rec may selectively influence some but not all mechanisms by an anti-tumor drug, that is, DTX might inhibit cellular mitosis mechanisms DTX [34] but not  modulate the DNA related metabolism by CRDCA [35] and PEM [36].”.

  1. the data are all still disconnected and observational. The new version includes a revised Table now spreading over 12 pages which still doesn’t address the issue raised which is that the RNAseq data doesn’t provide any explanation of the phenomena observed. I considered the initial manuscript to require major changes and this version seems to have had only cosmetic changes. To me, it does not warrant publication.

Answer; Thank you for these constructive comments, and we apologize for this misunderstanding of how we should properly answer these questions. Our intent was to honestly show several DEGs related to G-protein coupled receptor signaling, CREB signaling and Gia mediated signaling, which may presumably be involved in the Rec induced phenomena observed. Therefore, I was prepared such very busy Table 1. However, as pointed out, we completely agree that this Table 1 is difficult to read and may be only cosmetic changes, but we nevertheless feel that this information is still important because this Table includes important DEGs information forming the basis of three canonical pathways. In addition, information related to the method and results in terms of canonical pathway by IPA analysis was insufficiently described. Therefore, this Table 1 is included within the supplemental material, and more information related to the obtained three significant canonical pathway by IPA analysis with -log(p-value) were included in the method; “To predict possible upstream transcriptional regulators, DEGs were interpreted using the upstream regulator function of the ingenuity pathway analysis (IPA, Qiagen, https://www.qiagenbioinformatics.com/products/ingenuity-pathway-analysis) [33]. The significance of the biofunctions and the canonical pathways were evaluated by the Fisher Exact test p-value. Biofunctions were categorized as: Disease and Disorders; Molecular and Cellular Functions; and, Physiological System Development and Function. Alternatively, the canonical pathways were categorized as Metabolic Pathways and Signaling Pathways. Canonical pathways can also been ordered by the ratio (the number of molecules in a given pathway that meet cut criteria, divided by the total number of molecules that make up that pathway) [33]. “, and last paragraph of Results; “To elucidate the currently unidentified mechanisms responsible for inducing such characteristic features as above between A549 Rec and A549 MOCK, RNA sequence analyses were performed. As shown in MA and volcano plots (Fig. 6 A, B), 32 significantly up-regulated and 50 down-regulated differentially expressed genes (DEGs) were identified for A549 Rec as compared to A549 MOCK with a significance level of <0.05 (FDR) and an absolute fold-change ≥2 was identified (the list of the up-regulated and down-regulated genes is shown in supplemental Table 1). To estimate the possible functional roles of the above DEGs that were detected, we conducted a GO enrichment analysis and an Ingenuity Pathway Analysis (IPA) and (Qiagen, Redwood City, CA). The top three significant canonical pathways, G-protein coupled receptor (GPCR) signaling (-log(p-value) = 6.3), CREB signaling (-log(p-value) = 5.8), Gia mediated signaling (-log(p-value) = 5.2) were identified based upon the detected DEGs (supplemental Figs. 2-4). “.

Round 3

Reviewer 2 Report

the current version of this manuscript and the response by the authors is welcome. I accept that the full concentration-response curves are previously published and therefor a 3-point comparison is ok here. I would revert to the figure shown in the first version though. You can't have a figure 1 as presented in which the data points are connected by a smoothened curve. The original bar charts fit the new wording.

Presenting the long table as supplementary data is beneficial. There is lots of duplicated information in that table particularly in the Molecular Function Term and also the Biological Process Term columns. e.g. on the first entry for Recoverin itself there is duplication of "calcium ion binding" - I would think that table could still be tidied up a bit with removal of duplicate information.

Author Response

Dear Editor,

Thank you very much for the constructive comments concerning our manuscript, " G-protein coupled receptors mediated modulations of cell viability and drug sensitivity by the aberrant expressed recoverin within A549 cells”. We examined the Reviewer's comments carefully and prepared a revised version of our paper that takes these comments into account for resubmission. Therefore, we will greatly appreciate it if you will consider our revised paper for possible publication in IJMS. The changes are listed below. 

Reviewer 2

  1. the current version of this manuscript and the response by the authors is welcome. I accept that the full concentration-response curves are previously published and therefor a 3-point comparison is ok here. I would revert to the figure shown in the first version though. You can't have a figure 1 as presented in which the data points are connected by a smoothened curve. The original bar charts fit the new wording.

Answer; Thank you for this comment. As suggested, Fig.1 was reverted to the first version.

  1. Presenting the long table as supplementary data is beneficial. There is lots of duplicated information in that table particularly in the Molecular Function Term and also the Biological Process Term columns. e.g. on the first entry for Recoverin itself there is duplication of "calcium ion binding" - I would think that table could still be tidied up a bit with removal of duplicate information.

Answer; Thank you for this comment. As suggested, supplemental table was tidied up to avoid duplicate information.
